# ‘Nutritional Footprint’ in the Food, Meals and HoReCa Sectors: A Review

**DOI:** 10.3390/foods12020409

**Published:** 2023-01-15

**Authors:** Inmaculada Zarzo, Carla Soler, Maria-Angeles Fernandez-Zamudio, Tatiana Pina, Héctor Barco, Jose M. Soriano

**Affiliations:** 1Food & Health Lab, Institute of Materials Science, University of Valencia, 46980 Paterna, Valencia, Spain; 2University Clinic of Nutrition, Physical Activity and Physiotherapy, Lluís Alcanyís Foundation-University of Valencia, 46020 Valencia, Valencia, Spain; 3Joint Research Unit on Endocrinology, Nutrition and Clinical Dietetics, Health Research Institute La Fe-University of Valencia, 46026 Valencia, Valencia, Spain; 4Centro Para el Desarrollo de la Agricultura Sostenible, Instituto Valenciano de Investigaciones Agrarias, 46113 Moncada, Valencia, Spain; 5Department of Experimental and Social Sciences Education, University of Valencia, 46022 Valencia, Valencia, Spain; 6Enraíza Derechos, 20012 San Sebastian, Gipuzkoa, Spain

**Keywords:** nutritional footprint, nutritional tool, HoReCa sectors, sustainability, food, menus, meals

## Abstract

Nowadays, the food industry is integrating environmental, social, and health parameters to increase its sustainable impact. To do this, they are using new tools to calculate the potential efficiency of nutritional products with lower levels of environmental impact. One of these tools is called the ‘nutritional footprint’, created by Wuppertal Institute for Climate, Environment and Energy GmbH. This study aims to review this concept and clarify its historical development, its use in several sectors of the food industry, and its transformation from a manual to an online tool. Results reflected that it is a suitable indicator that integrates nutritional, environmental, and social-economic dimensions to help the decision-making process in the procurement of more sustainable products and, although it is limited to Germany due to the use of the national standard nutritional intakes of Germany, its importance lies in the fact that is a promising instrument to promote environmental sustainability in the context of food, meals, and the hotel, restaurant and catering (HoReCa) sectors.

## 1. Introduction

The 2030 Agenda for Sustainable Development, including the 17 Sustainable Development Goals (SDGs) and their 169 targets, through their holistic and universal approach, are adapted to different contexts by connecting scientists with companies, governments and society, and, from an inter- and transdisciplinary approach, are applied in several industries, including agri-food [1] and the hotel, restaurant, and catering (HoReCa) [2] sectors. In recent years, the search for meals or dishes, according to current nutritional recommendations and environmental [3] aspects, has increased either individually or jointly with these or other indicators, such as cultural [4], social [5], political [6], and/or economic indicators [7]. In fact, the FAO [8] justified that the sustainability of diets should focus on the nutritional, environmental, economic and socio-cultural dimensions. Grigoriadis et al. [9] indicated that these diets can efficiently contribute to achieving nutrient-adequate dietary patterns and, at the same time, fewer contaminants. In fact, these previous authors justified the necessity of working towards a combined measure for describing the environmental impact and nutritive value of foods grouped in four approaches: programming optimization, statistical analysis, graphical representation, and indices. To help in the evaluation of the ‘sustainable assessment’ of foods [10], several instruments and methods have been developed considering: (i) health, such as ‘conscious enjoyment’, ‘healthy meal index’ and the WHO/European pledge instruments of ZFV companies [11], the Technical University of Denmark [12] and WHO Europe [13], respectively; (ii) the environment, including the ‘Eaternity app’ and BEELONG by Eaternity [14] and Ecole hôtelière de Lausanne [15], respectively; and (iii) both previous indicators, such as the ‘Menu Sustainability Index’ (MSI), SusDISH and the ‘nutritional footprint’ by Zurich University of Applied Sciences, together with ZFV [16], Martin Luther University Halle-Wittenberg [17] and Wuppertal Institute for Climate, Environment and Energy GmbH [18], respectively. For combined health–environmental tools, MSI was used, which rates meals on the basis of nutritional balance and environmental friendliness separately, but does not take beverages into account [16]. SusDISH was developed as a software-based concept, including 16 health indicators and 15 ecological indicators (where a life-cycle-based concept of ecological shortage is used) [17]. The ‘nutritional footprint’ combines the two dimensions of environment and health into a single index [18]. According to this latter concept, the Wuppertal Institute developed the project and application called ‘development, testing, and distribution of concepts for sustainable production and consumption in the field of out-of-home catering (NAHGAST)’ [19]. Abbade [20] indicated that improvement is still needed, and there is also confusion with the term, which can be either ‘nutritional footprint’ [21] or ‘nutrition footprint’ [22]. The aim of this work is to review the concept of a ‘nutritional footprint’ and the NAHGAST project in food, meals, and the HoReCa sectors, to observe the evolution in literature since its inception and to study the applicability in these sectors.

## 2. Review Method

The key question to be answered in this manuscript is the usefulness of the concept of the ‘nutritional footprint’ and NAHGAST for the food, catering, and HoReCa sectors as sustainable, manual, and automated food tools, respectively, that help to link health, the economy, and society and nutritional aspects in a unique index. This review was carried out with the Preferred Reporting Items for Systematic Reviews and Meta-Analyses (PRISMA) statement [23] (Figure 1) using the three databases Web of Science, PubMed, and Scopus. The Boolean strings chosen were (‘nutritional footprint’) AND (restaurant OR ‘food industry’ OR catering OR meal OR menu OR food OR lunch OR dish OR recipe OR HoReCa). EndNote software, version X7.0 (Thomson Reuters, New York, NY, USA) was used to find duplicate articles by screening titles and abstracts. 

The searches included works published in all languages and restricted to the last 20 years (from 1 November 2002) and were carried out with a comprehensive search of three types of literature (‘white’, ‘grey’, and ‘black’), applying the use of four different search tools, such as (i) grey literature databases; (ii) personalization of Google search engines; (iii) specific websites; and (iv) advice with specialists contacted.

Letters, editorials, comments, unpublished data, other types of non-peer-reviewed publications and articles without full texts or not related to the ‘nutritional footprint’ indicator applied to sustainability were excluded.

Two teams of matched reviewers (I.Z., M.A.F.Z., T.P., J.M.S.) experienced in systematic reviews independently screened titles, abstracts and full texts for eligibility, assessed generalizability and collected data from each article. The third team of paired reviewers (C.S., H.B.) helped to resolve any disagreements. Two teams evaluated the risk of bias using the Critical Appraisal Skills Programme (CASP) tool (http://www.casp-uk.net/ (accessed on 1 January 2023)), which graded each study based on several components, such as the appropriateness of the study design for the research question, the risk of selection bias, exposure and outcome assessment. Furthermore, the presence of discrepancies was solved by the third team reviewer. From eligible studies, collected data were extracted and grouped with the following variables: sample, health, and environmental indicators, year of publication, country, value, or range of ‘nutritional footprint’ and author. Results were grouped into two sections: ‘nutritional footprint’ and NAHGAST. Finally, 94 articles had to be excluded from this manuscript for not meeting the inclusion-exclusion criteria, which was 17 references obtained, to work in this systematic review. Furthermore, almost all selected articles (16 of 17 studies) were evaluated as having a ‘moderate’ risk of bias.

## 3. ‘Nutritional Footprint’

As a curiosity, ‘nutritional footprint’ was first used by an American manufacturer and retailer of outdoor footwear, Timberland LLC, being used on its shoeboxes to provide information about its environmental and community impacts [24], but this had nothing to do with the nutritional topic. The first reference applied in health literature was by Sanhuenza and Valenzuela [25] to create a personalized diet according to the genetics and/or phenotype of individuals. On the other hand, the concept of ‘maternal nutritional footprint’ has been directed, to avoid confusion, toward the concept of Developmental Origins of Health and Diseases (DOHaD) [26], which explains how the maternal environment affects fetal development due to hormonal imbalance, oxidative stress and epigenetic modifications. Neither of the latter articles consider the nexus of nutrition with the environment or other aspects, such as the social and economic dimensions. On the other hand, 16 references were closely linked to nutritional and environmental aspects in the food industry, all of them being from menus, dishes, or foods served in the HoReCa sector. The first reference to a ‘nutritional footprint’ in the food industry was not made until 2013 and was developed by The Wuppertal Institute [27]. This research group justified that the indicators used for the development of this concept were based on Liedtke’s study [28], which included two combined types of indicators to study the effect of diets on health (energy value and nutrient density) and the environmental impact and resource intensity of diets (abiotic and biotic compounds, erosion and land displacement, water and energy consumption, use and land yield, biodiversity, and CO_2_ equivalents). Table 1 shows the characteristics and main findings of some selected articles in the present review with a range of values of the ‘nutritional footprint’ reflected only in 8 out of 17 articles selected for this review. Lukas et al. [27] indicated that the established reference values correspond to the sum of both indicators ranging from 2 (environmental impact = 1 + health impact = 1) to 6 (environmental impact = 3 + health impact = 3) reflecting a ‘good’ or ‘worse’ product with little or high impact, respectively, on health and the environment. This study evaluated two foods, lettuce and beef, the latter having the highest value (4.9 versus 2.3). In the same year, these authors [29] carried out a scientific revision to decrease the number of indicators based on international and German agencies [30,31,32] for health indicators (calorie (kcal), salt (g), dietary fiber (g) and saturates (g) contents) and several studies [33,34] on environmental indicators (material (g), carbon (g), and water (L) footprints and land use (m^2^)), illustrating the whole value chain and its direct and further indirect effects. Reference values ranged from 1 to 3 for a minor and major ‘nutritional footprint’, respectively, using this formula: 

‘Nutritional footprint’ = (environmental indicators + health indicators)/2

Lukas et al. [29] calculated the ‘nutritional footprint’ using both of these indicators in a classic restaurant dish (burger with a large portion of French fries (160 g) with mayonnaise (20 g) and a 500 mL soft drink) versus a fitness dish (‘Italian chicken mozzarella’ wrap with classic snack salad with balsamic classic dressing, a portion of organic apples (80 g) and 500 mL water), obtaining ‘nutritional footprint’ values of 2.34 and 1.58, respectively. Lukas et al. [35,36] used their parameter on five menus: (i) burger (burger with double beef patty, with chips and cola); (ii) chili without meat (vegan chili without meat with white bread and apple juice with mineral water); (iii) lasagna (vegetarian lasagna with a salad side dish and apple juice with mineral water); (iv) meat roll (meat roll with red cabbage side dish, potatoes and water); and (v) wrap (wrap with chicken, with a salad side dish, apple and water), resulting in values of 2.625, 1.125, 1.25, 2.625, and 1.375, respectively. Three and two of them had a low- and high-quality ranking, according to a value of less than 1.65 (low negative effect on the environment or health), between 1.65 and 2.3 (intermediate effect), and above 2.3 (strong effect). Lukas et al. [37] restored three levels of the ‘nutritional footprint’ with a new classification as ‘low’, ‘medium’, and ‘high’ effects, ranging from 1 to 1.6, 1.6 to 2.2, and >2.2, respectively, which was applied to eight German lunch menus: (i) spaghetti Bolognese salad; (ii) sausage with curry and chips with mayonnaise; (iii) roll with beef in red wine sauce, vegetables, and potatoes; (iv) large mixed salad with baguette; (v) breaded fish fillet accompanied by a remoulade sauce, broccoli and potatoes; (vi) vegetable lasagna; (vii) chili without meat and with bread; and (viii) potato pancake with applesauce, with values of 2.25, 2.25, 2.5, 1.125, 1.75, 1.25, 1.125, and 1.5, respectively. Müller et al. [10] indicated that MSI showed the efficacy of the separate use of environment and health indicators, due to the fact that the consumers selected dishes based on personal preference, catering staff, and convenience. In our viewpoint, we do not agree with this idea due to the fact that, in the last five years, users, consumers, and policymakers have raised awareness of the environment-health binomial throughout the world.

On the other hand, the ‘nutritional footprint’ [21] should not be confused with ‘nutrition footprint’. This latter term has been used by Grönman et al. [41] as an indicator during the whole life cycle of a product, being annexed with nutrient (nitrogen, phosphorus, and potassium) intake and nutrient-use efficiency, and extrapolated in the Finnish beef production and consumption chain by Joensuu et al. [22]. 

## 4. ‘Development, Testing and the Distribution of Concepts for Sustainable Production and Consumption in the Field of Out-of-Home Catering (NAHGAST)’

The studies reflected in the previous section are part of the NAHGAST project [42] and were applied to the beef roll menu, obtaining a ‘nutritional footprint’ value of 2.5. Bienge et al. [43] carried out the relationship of NAHGAST indicators with Sustainable Development Goals (SDGs). The manual use of this tool derived from the free online creation (https://www.nahgast.de/rechner/ (accessed on 1 January 2023)) with financing from the German Federal Ministry of Education and Research and known as the NAHGAST I project, launched in March 2018. This online software included around 1500 meals. Furthermore, the use of NAHGAST I is preferable in places where only three main dishes were served, to compare information among intervention weeks and cafeterias [44]. On the other hand, the aim of the interventions in NAHGAST I was to make guests of out-of-home catering aware of the most sustainable daily dishes with small ‘nudges’, so that they choose them more often. System 1 nudges that are more subconscious (change in the serving position, change in the menu position, descriptive names) were used here, as well as system 2 nudges that stimulate thought processes and thus have a more cognitive effect (labels and information) [45]. As a result of this tool, around 74,000 dishes from the harmonized menus were analyzed, indicating that the nudge of the change in the serving position showed a significantly positive effect (+22.5% increase in sales compared to the baseline measurement). The change in the menu position (+3.5%) reflected a positive effect on the sales figures, but not significantly. All other nudges (descriptive names, labels and information) had almost no influence on the sales behavior of the guests or even had a negative effect [46]. In addition, there were strong sales deviations between the individual partners and between the most sustainable dishes of the day. These studies reflected that not only the choice of sustainable food but also the consumption of the chosen food is relevant for the sustainability of a selection decision. For this reason, the application of this procedure should pay special attention to the topic of food waste when used in living laboratories to carry out interventions to reduce leftover plates/food waste [47].

In 2020, NAHGAST II, helped by 20 practical partners, was presented to society, allowing connectivity in several digital systems, such as recipe, accounting and resource management [39]. This second version demonstrated that the first version of NAHGAST could be classified as comprehensive and most representative. In fact, its use in the out-of-home consumption sector, other food companies or even for customers in out-of-home and private households, demonstrated efficiency to verify the reduction in the environmental parameters, i.e., changing beef to soy in dishes reduced the carbon and material footprint by around 34% and 35%, respectively. NAHGAST I demonstrated that individual nudges were able to have a positive effect, while others did not show the desired effects and the follow-up project NAHGAST II, therefore, took up this topic again [48]. On the other hand, two modules, NAHGAST Meal-Basic and NAHGAST Meal-Pro, are included in the NAHGAST assessment procedure and consist of 12 (6 ecological (percentages of sustainably caught fish and percentages of animal, seasonal, regional, organic products and GMO-free products), 1 social (percentages of fair-trade products), 3 health (percentages of fruits and vegetables, energy and fiber) and 2 economic (popularity and cost-coverage) parameters) indicators and 14 (4 ecological (material and carbon footprint, water demand and area required), 2 social (fair trade and animal welfare), 6 health (energy, fiber, fat, carbohydrates, sugar, salt) and 2 economic (popularity and cost-coverage) parameters) indicators, respectively [19]. The score scale was grouped as ‘not recommendable’, ‘restrictively recommendable’ and ‘recommendable’ for values situated from 1 to 1.4, from 1.5 to 2.4 and from 2.5 to 3, respectively, and visual information was used for them in red, yellow, and green, respectively. Engelmann et al. [49] indicated that the Meal-Basic version is useful and easy to control by kitchen workers. The NAHGAST project received The KlimaExpo.NRW award, recognizing the NAHGAST research project for its exemplary commitment to climate protection. As a summary of both versions, Figure 2 shows the difference between NAHGAST Meal-Basic and NAHGAST Meal-Pro.

Speck et al. [50,51] indicated that several parameters, such as popularity and cost recovery, both without quantified target value, for the economic parameter, were studied but they were not incorporated into the final implementation of this online tool. In this research, they examined the nutritional and environmental approach to food in three different settings: (i) review of recipes, in public catering companies, without the help of experts; (ii) review of recipes aligned with energy-nutritional recommendations [52,53] with products of animal origin (<30%) in the menu composition; and (iii) recipe revision aligned with established guidelines for the carbon (<800 g) and material (<2670 g) footprints of this studied meal. Their results demonstrated significantly that a recipe revision without scientific support and/or the implementation of recommended intakes affected the values in the reduction in greenhouse gas emissions and the use of natural resources applied to business catering, but the value doubled if the target values of environmental indicators and their implementation were more rigorously focused. Furthermore, not all scenarios implied a deterioration of the considered nutritional parameters, and optimization using the public catering companies themselves could save, yearly, in business catering, around 1.3 and 0.3 million tons of resources and greenhouse gas emissions, respectively. The same authors [39] carried out another study where nutritional values, including energy (<670 kcal/<830 kcal), fat (<24 g/<30 g), carbohydrates (<90 g/<95 g), sugar (<17 g/<19 g) and fibers (>8 g/>6 g), environmental values, including material (<2670 g/<4000 g) and carbon (<800 g/<1200 g) footprints, water (<640 L/<975 L) and land use (<1.25 m^2^/<1.875 m^2^), and social values, including partially fairtrade ingredients (>90%/>85%) and partially animal-based food that fosters animal welfare (>60%/>55%) indicators, were used to calculate the nutritional footprint of 1509 meals. Their results reflected that the values of those three diets were high for the vegan diet (4.71), followed by vegetarian meals (4.53) and, finally, by a mixed diet (3.86). Recently, Wuppertal Institute has been developing the BiTe (Biodiversität über den Tellerrand) project, whereby they want to integrate into the database of the existing NAHGAST calculator the biodiversity index for measuring (agro)biodiversity to make the influence of a lunchtime meal on (agro)biodiversity measurable using a co-creation process with various companies from the field of out-of-home gastronomy [54].

The COVID-19 pandemic established a new strategic framework in food, meals and HoReCa sectors, which is the 6S concept (food safety for food security, scientific, sovereignty, solidarity-based, and sustainable) [55,56,57], as the new paradigm that should be implemented in these sectors. The ‘nutritional footprint’ and NAHGAST were the prelude to a new focus that redirected to a new green circular economy based in the Society 5.0 era [58], in which individuals take the initiative in health management, helped by digital innovation with artificial intelligence, the Internet of Things and other digital technologies and big data. NAHGAST was developed in this direction and was ahead of its time, but the way to achieve this new era is to harmonize the 6S concept and contribute to SDGs. Let us not forget that, today, health and sustainability are an inseparable combination for quality of life and the planet. Gupta et al. [59] carried out a systematic review of reviews and evidenced that the implementation of interventions to help food retail environments promote health requires targeting an individual–intrapersonal–environmental trinomial and your idea, as a starting point in food retail, to help build research and practice partnerships that support business and health, and can be applied in the food, meals, and HoReCa sectors. However, the efficacy of this latter point should be helped by environmental literacy and food education simultaneously, as is demonstrated in China [60], Europe [61], Australia [62], and the USA [63], among others. 

## 5. Conclusions and Future Directions

In our viewpoint, the main conclusion reflected that the studied footprint has evolved over time and could help to work with a tool that considers the nexus of nutrition with the environment or other aspects, such as the social and economic dimensions, but it is limited to the national standard dietary intake of Germany [30], meaning that any benchmarking with other countries is flawed. Additionally, the term ‘nutritional footprint’ is not a registered name, which could mislead researchers and consumers. In fact, Arrieta and González [64], in 2019, also defined a ‘nutritional footprint’ indicator, which helped derive energy use and greenhouse gas emissions represented per unit of nutritional content rather than per unit of weight, but was related to protein. They did not relate with the term of Lukas et al. [27,29], but included the total footprint (from cradle to table) focusing on the preparation of the 18 samples from meat, cereal, and legume groups, demonstrating that the highest efficiency for energy (30.1 g of protein per MJ) and greenhouse gas emissions (458 g protein per kg CO_2_-eq. emitted) was obtained with soybeans. Irrespective of this, ‘nutritional footprint’ could be considered an important factor reflected by the FAO [65], which emphasized the term ‘nexus’ in nutritional, environmental, economic, and socio-cultural indicators, reflecting that action in one of the dimensions has implications on the others, enhancing synergies and trade-offs to help possible answers focused on environmental sustainability. New searches are underway to increase other tools, such as the integrated Water–Energy–Food–Climate Nexus Index (WEFCNI) developed in the anchovy canning industry in northern Spain by Laso et al. [66]. This tool was applied, four years later, in the European Atlantic Area NEPTUNUS project, which focused on the sustainability of the seafood sector, with the aim of achieving a circular economy through the use of eco-innovation and eco-labelling of products [67]. In our viewpoint, the future footprint index should attach equal importance to health (nutritional), environmental, economic, and socio-cultural parameters to harmonize those indicators with global values, including international reference intakes. In the end, the ultimate goal of this future tool, non-existent at present, is that consumers enjoy eating and the work of catering staff is affordable and cost-effective to run, in addition to safeguarding the planet.

## Figures and Tables

**Figure 1 foods-12-00409-f001:**
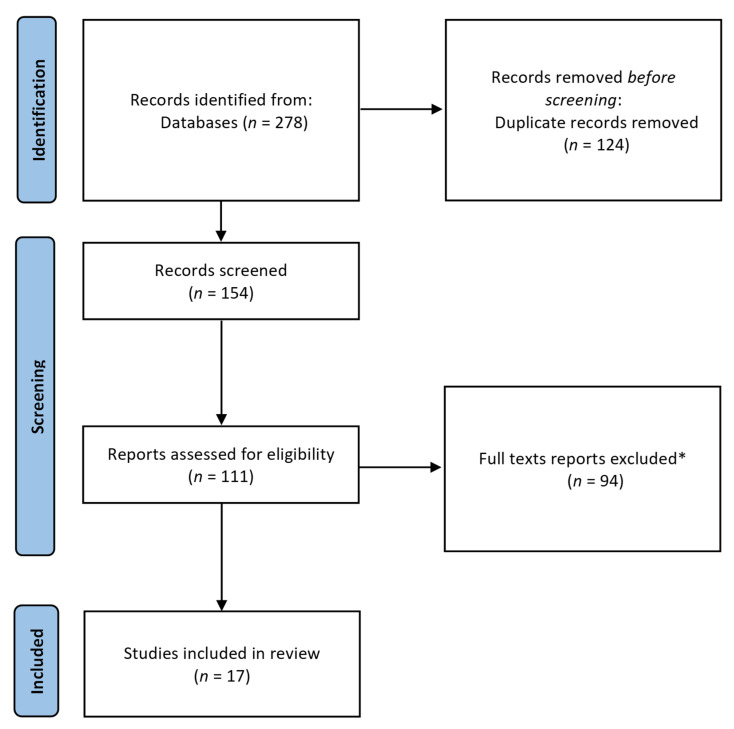
Flow diagram, according to Preferred Reporting Items for Systematic Reviews and Meta-Analyses (PRISMA) [23], used in this study. * Exclusion criteria are reflected in the ‘review method’ section.

**Figure 2 foods-12-00409-f002:**
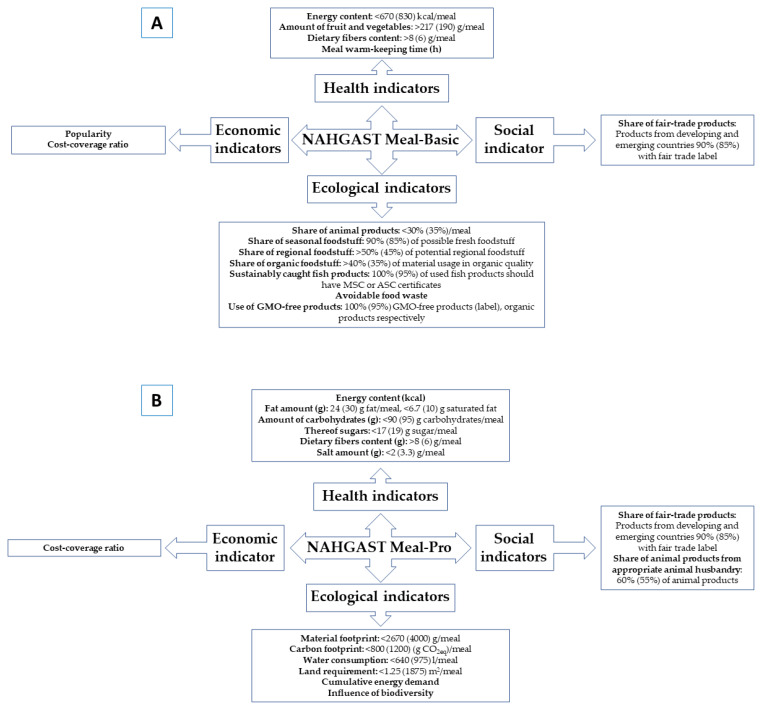
Differences between NAHGAST Meal-Basic (**A**) and NAHGAST Meal-Pro (**B**).

**Table 1 foods-12-00409-t001:** Some values/ranges of the nutritional footprint of different foods, menus, and meals.

Sample	Health Indicators	Environmental Indicators	Year	Country	Value/range of Nutritional Footprint	Ref.
Lettuce and beef	Nutrient density and calorie content	Abiotic and biotic material, erosion and earth movement, water consumption, energy consumption, land use and yield, biodiversity and CO_2_-equivalents	2013	Germany	2.3 and 4.9	[27]
Two menus	Calorie intake (kcal), sodium (g), dietary fiber (g) and saturates (g)	Material ^a^ (g) and carbon (g) footprints, water use (L) and land use (m^2^)	2013	Germany	1.58 and 2.34	[29]
Five menus	Calorie intake (kcal), sodium (g), dietary fiber (g) and saturates (g)	Material ^a^ (g) and carbon (g) footprints, water use (L) and land use (m^2^)	2014	Germany	1.125–2.625	[35,36]
Beef roll menu	Calorie intake (kcal), sodium (g), dietary fiber (g) and saturates (g)	Material ^a^ (g) and carbon (g) footprints, water use (L) and land use (m^2^)	2015	Germany	2.5	[37]
Eight German lunch meals	Calorie intake (kcal), sodium (g), dietary fiber (g) and saturates (g)	Material ^a^ (g) and carbon (g) footprints, water use (L) and land use (m^2^)	2016	Germany	1.125–2.5	[38]
Beef goulash ^b^	Energy content (kcal), amount of fruit and vegetables (g), dietary fibers content (g) and time keeping meal warm (h)	Ecological (share of animal, seasonal, regional, organic, and sustainably caught fish products, avoidable food waste and use of GMO-free products), social (share of fair-trade products), economic (popularity and cost-coverage ratio) indicators	2018	Germany	1.6	[19]
Beef goulash ^c^	Energy content (kcal), fat amount (g), amount of carbohydrates (g) thereof sugars, dietary fibers content (g) and salt amount (g)	Ecological (material and carbon footprints, water consumption, land requirement, cumulative energy demand and influence of biodiversity), social (share of fair-trade products and share of animal products from appropriate animal husbandry), economic (cost-coverage ratio) indicators	2018	Germany	1.8	[19]
1509 meals	Energy content (kcal), fat (g), carbohydrates (g), sugar (g) and fibers (g)	Environment (material and carbon footprints, water use and land use) and social (share of fair-trade ingredients and of animal-based food that fosters animal welfare)	2020	Germany	3.86–4.71	[39]

^a^ The material footprint is the sum of all necessary resources needed during the complete life cycle of a product [40]; ^b^ using NAHGAST Meal-Basic; ^c^ using NAHGAST Meal-Pro.

## Data Availability

Not applicable.

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
