# Peer review of "‘Nutritional Footprint’ in the Food, Meals and HoReCa Sectors: A Review"

_foods, 2023, doi:10.3390/foods12020409_

Round 1

Reviewer 1 Report

Systematic reviews are important tools in providing evidence.

In a critical evaluation that systematic review, it is necessary to clearly define the question to be answered, in addition to how the methodological quality of the included studies was evaluated, and how the collected data were extracted.

The authors must be included how conducting risk of bias assessments and which software for find duplicate studies. 

Improve the introduction: describe and conceptualize the nutritional footprint detailing how to obtain the footprint value, how to use data from health indicators and environmental indicators for the formula. Perhaps a figure illustrating the calculation.

It was not clear in the results how to obtain the nutritional footprint value. Please, explore the methodology described in selected articles.

In the table 1, it is important that the 17 selected studies are described.

The title can be improved, emphasis of the study is more on the food, meals and Horeca sector than the food industry.

Reviewer 2 Report

Dear Authors, thanks for providing this study which summarises quite well the advancements of the literature on nutritional footprint. The topic is certainly up to date.

While I feel the study does a good job in reporting the history of the concept of nutritional footprint (although with some flaws, listed below), I am not sure it gives a significant contribution for the advancement of this stream of research.

I have some remarks about the methodology of the literature review:

- If the concept of ‘nutritional footprint’ was only established in 2013, why does the review go back until 2022? Did you select any article published before 2013 in the 11 considered for the review?

- in my view, the literature review is too much focused on the NAHGAST approach; if the paper wants to discuss the performance of this approach against others, it would be wise to give more space to other approaches as well

- I would have expected to read more about the exclusion criteria of the papers during the PRISMA methodology applied to the literature review

I also have some remarks about the way the results are presented and discussed; besides reporting the results of the papers included in the review, I would have expected to see more discussion of these results, in such a way that the paper could contribute to the stream of research.

For example, I would have expected a structured comparison between the NAHGAST approach and other alternative approaches, and a discussion about the pros and cons of each approach.

As for section 4, the discussion about the efficacy of the different nudging actions is very interesting; wouldn’t it be worth to summarise it with a figure or diagram? This would also help to draw, at the end of the discussion or in the conclusions, the priority for research on the nutritional footprint, which are currently too general in my opinion.

Round 2

Reviewer 2 Report

I thank the authors for the good review they have performed. 

In Figure 2, I suggest to refer to "economic indicators" rather than "economical indicators"

I acknowledge that you have added the objective of the review in the first part of the paper, but it sounds still too general to me. "The key question to be answered in this manuscript is if the concept of ‘nutritional footprint’ and NAHGAST is useful [...]" but useful to do what?

As a consequence of the objective being quite vague, the conclusions are not concrete enough in my opinion. To whom are they addressed? What is the key message emerging from this paper?

The main conclusion is that "the studied footprint in this review represents great progress and reflects the importance of having access to a tool focused on sustainability and healthy diet". I am not sure how this statement emerges from the review. 
